# Research and Development of Novel Refractory of MgO Doped with ZrO_2_ Nanoparticles for Copper Slag Resistance

**DOI:** 10.3390/ma14092277

**Published:** 2021-04-28

**Authors:** Cristian Gómez-Rodríguez, Yanet Antonio-Zárate, Josept Revuelta-Acosta, Luis Felipe Verdeja, Daniel Fernández-González, Jesús Fernando López-Perales, Edén Amaral Rodríguez-Castellanos, Linda Viviana García-Quiñonez, Guadalupe Alan Castillo-Rodríguez

**Affiliations:** 1Faculty of Engineering, University of Veracruz (Coatzacoalcos), Av. Universidad km 7.5 Col. Santa Isabel, Coatzacoalcos 96535, Mexico; crisgomez@uv.mx (C.G.-R.); yantonio@uv.mx (Y.A.-Z.); jrevuelta@uv.mx (J.R.-A.); 2Department of Materials Science and Metallurgical Engineering, University of Oviedo, 33003 Oviedo/Uviéu, Spain; lfv@uniovi.es (L.F.V.); fernandezgdaniel90@gmail.com (D.F.-G.); 3Faculty of Mechanical and Electrical Engineering, Autonomous University of Nuevo Leon, San Nicolás de los Garza 66450, Mexico; jlopezp@uanl.edu.mx (J.F.L.-P.); eden.rodriguezcs@uanl.edu.mx (E.A.R.-C.); 4Catedras-CONACYT, Center Scientific Research and Higher Education of Ensenada (CICESE) Monterrey, Apodaca 66603, Mexico

**Keywords:** ceramic composites, corrosion, zirconia, nanoparticles, MgO refractories

## Abstract

This study investigates the corrosion mechanism on 100 wt.% MgO and 95 wt.% MgO with 5 wt.% nano-ZrO_2_ ceramic composites. First, MgO powder and powder mixtures (MgO + nano ZrO_2_) were uniaxially and isostatically pressed; then, they were sintered at 1650 °C. Corrosion by copper slag was studied in sintered samples. Physical properties, microstructure, and penetration of the slag in the refractory were studied. Results reveal that ZrO_2_ nanoparticles enhanced the samples’ densification, promoting grain growth due to diffusion of vacancies during the sintering process. Additionally, magnesia bricks were severely corroded, if compared with those doped with nano-ZrO_2_, mainly due to the dissolution of MgO grains during the chemical attack by copper slag.

## 1. Introduction

The copper industry is a great consumer of refractory materials since more than 25,000 t of refractory materials are used in its primary production (copper smelting, converting, and refining furnaces). It has been estimated in 35 Mt of copper worldwide produced every year: 60% by pyrometallurgical processes and 40% from copper scrap [1].

Copper can be produced through different technologies depending on the ore. Copper sulfide is known as the principal copper ores, which includes chalcopyrite (CuFeS_2_), bornite (Cu_5_FeS_4_), and chalcocite (Cu_2_S). The fusion-conversion process is the most widely used in copper metallurgy. The fusion process generates two immiscible phases: matte (copper as a sulfide) and slag (iron as fayalite). Copper in the slag will be as sulfide (matte), which is dragged or trapped by the slag, or as oxide associated with other slag oxides of the slag. Copper slags are disposed of in dumps or sold as abrasives or gravel for mortar [2] when the copper content is <2%. Copper slags are formed mainly by fayalite, magnetite, and copper trapped as oxide or sulfide in quantities between 0.5 and 2% depending on the process. Apart from the copper content, copper slags are characterized by their high iron content (>40%, [3,4,5,6,7,8,9]). The refractory bricks essentially used to line nearly all of the smelting, converting, and refining furnaces used for the production and refining of molten copper are magnesia-chrome refractories. As it is known, matte and slag phases attack the refractory lining at a high temperature in a detrimental way. Matte and slag phases increase the deterioration in the slag line [10]. These drawbacks make the copper industry acquire such quantities of refractory materials for the total or partial replacement of damaged refractory bricks from industrial equipment or furnaces [11].

In this context, continuous efforts are made by groups of researchers to improve the lifetime of refractory linings of various copper-making furnaces via lining concept optimization. The most feasible way to enhance the performance of furnace linings is through the development of new refractory. These refractories must have exceptional properties against the corrosion phenomenon caused during the coopering process. Therefore, the mechanisms of chemical degradation of magnesia-chromite refractory have been studied widely [10,11].

Chen et al. studied post-mortem magnesia-chromite refractory samples from copper smelting furnaces. They were compared with fresh magnesia-chromite refractory, tested with industrial copper smelting slag and Cu_2_O at laboratory study. They concluded that CuO, oxidized from matte, can severely react with magnesia to form the guggenite phase. This phase affects the refractory [12].

Yibiao Xu et al. studied the effect of Al_2_O_3_ and TiO_2_ on the properties of a magnesia-chrome refractory. They found that the corrosion resistance decreases slightly with the Cr_2_O_3_ content due to the increased apparent porosity, which would result in the penetration of more corrosive copper slag. Newly formed (Mg, Cu) (Cr, Fe)_2_O_4_ spinel dense layer between the slag and penetration layer effectively hinders further penetration of slag and well protects the specimens. TiO_2_ addition can suppress the formation of Cr (VI) effectively [13].

The investigation focused on the use of refractories containing magnesium–aluminum spinel (MgAl_2_O_4_) as a replacement for magnesia-chrome brick; the choice of this material stems from the similarity of the MgO–Al_2_O_3_ and MgO–Cr_2_O_3_ phase diagrams and the use of spinel-containing refractories as a replacement for magnesia-chrome in other industries. As a result, it appears that the solution to this problem will come from refractories in systems other than MgO–Al_2_O_3_ since it reacts with copper slag [14].

The degradation mechanisms of alumina–chromite refractories in a Cu-Cu_x_O mixture and a secondary copper smelting slag were investigated. As a result, alumina in the alumina-chromite refractory is corroded by the slag. The Cr_2_O_3_-rich phase in the refractory is little attacked. The high solubility limit of Al_2_O_3_ in the slag is the thermodynamic driving force for the faster dissolution of the alumina grains. The slag infiltrated into the alumina-chromite refractory through the open-pore network and the grain boundaries, and by the corrosion of the CaO·6Al_2_O_3_ impurity phases. The infiltration of Cu–Cu_x_O mixture into the refractory via the open-pore network and the grain boundaries was limited by spinel formation. (Fe, Mg, Zn)(Al, Cr)_2_O_4_ spinel phases are generated from the slag/refractory interactions at the refractory surface and in the infiltrated refractory. The spinel was formed next to the infiltrating slag and filled the open pores and cracks in the refractory, leading to a densified refractory zone [15].

Q. Zhen et al. studied the effect of the addition of MgAl_2_O_4_ spinel powder on the physicochemical properties and anti-slag of the alumina-chrome refractory. MgAl_2_O_4_ spinel as an additive improved the slag corrosion and penetration resistance of the alumina-chrome refractory by 33.3% and crude copper corrosion and penetration resistance by 42.8%, owing to the lower porosity and generation of Mg(Fe, Al)O_4_ [16].

Volker Stain et al. studied new bricks based on the systems: sintered magnesia-SiC, fused magnesia-graphite, and olivine-graphite. They concluded that these refractories show a similar wear rate as magnesia-chromite bricks but have an infiltration barrier. The olivine formation between slag-fayalite and brick-forsterite in the reaction zone formed a semi-frozen layer at the interface that interrupts infiltration [17].

MgO-Chrome refractory family is characterized by each other due to their type of bond, composition, purity level, and meet the demands on properties that the copper industry requires [11]. However, during the smelting and converting processes, Cr^3+^ may oxidize to Cr^6+^, causing environmental problems [18]. For this reason, other chromium-free materials have been studied as prospects in the substitution of MgO-Chrome.

Additionally, magnesia grains react directly with SiO_2_ from slag to form forsterite, while the formation of an accretion layer consumes the refractory [19]. In other work, the effect of ZnO level in secondary copper smelter slags on magnesia-chromite refractories was studied by Chen et al. It was concluded that the dissolution of periclase in the fayalite slag containing ZnO is not improved by increasing the ZnO content in slags [20].

On the other hand, nanotechnology has been very active in the refractory field to improve properties. It has been reported that nanoparticles increase the density, induce the early formation of phases with high melting point, and improve the corrosion, spalling, and oxidation resistance on refractory materials [21]. Remarkable results using nanometric materials such as Fe_2_O_3_ [22,23], Al_2_O_3_ [23,24,25], ZrO_2_ [26,27], ZrSiO_4_ [28,29], C [30,31], MgAl_2_O_4_ [32], TiO_2_ [33,34], Cr_2_O_3_ [35,36], SiO_2_ [37] in the MgO matrix have been reported. Nanometrics materials permit to achieve properties (physical, mechanical, chemical, and thermal) superior to the conventional ones. In this sense, nano-ZrO_2_ has proved to help in the MgO sintering mechanism. In addition to being more reactive than ZrO_2_ microparticles and forming the CaZrO_3_ phase, which is a highly refractory phase in basic slag environments [38].

According to the chrome-free refractory concept, this research aims to study the corrosion resistance of a MgO-based refractory with 5% nano-ZrO_2_ addition in contact with copper slag. The main objective is the development of a new chrome-free value-added product (i.e., MgO-based refractory with 5% nano-ZrO_2_ addition) with promising application in the copper industry to meet the chemical degradation of molten phases, abrasion resulting from the movement of the bath inside the furnaces, thermal, and mechanical shocks [10].

## 2. Materials and Methods

MgO (Industrias Magnelec SA de CV, Saltillo, Coah, Mexico) with particle size <45 µm and chemical composition (wt.%): SiO_2_, 0.2; Al_2_O_3_, 0.15; Fe_2_O_3_, 0.13; MgO, 98.5; CaO, 1; B_2_O_3_, 0.01; was used in the experiments. High purity nano-ZrO_2_ with particle size <80 nm (Sigma-Aldrich, Inc., St. Louis, MO, USA) was also used as raw material. A dispersion of acetone and anionic polymeric dispersant (Zephrym-PD3315, Croda Industrial Chemicals, Snaith, UK) was used to prepare the green specimens. Samples were cold-pressed to obtain the green specimen: first, uniaxially pressed (UP) in a steel mold at 100 MPa for 2 min and, then isostatically pressed (IP) in an autoclave (Autoclave Engineers, Inc P-419 (located in Centro de Investigacion en Nanoteriales y Nanotecnologia (CINN), el Entrego, Asturias, Spain) at 200 MPa for 5 min. Finally, green specimens were sintered at 1650 °C for 4 h. The control of grain size, agglomeration, pressing processes, and the sintering temperature are designed to minimize porosity in the final refractory brick. Physical properties were evaluated in terms of apparent porosity (AP) and bulk density (BD) by the Archimedes method. Mechanical characterization was determined in terms of cold crushing strength (CCS method). A universal testing machine (ELE-International, ABR-AUTO model, located in Universidad Autonoma de Nuevo Leon, San Nicolas de los Garza, Nuevo Leon, Mexico) was used. Reported values are the average of 10 determinations for each composition. The crystallographic phases and microstructural characteristics were evaluated by X-ray diffraction (XRD) and scanning electron microscopy (SEM), respectively.

Table 1 shows the values of density, porosity, mechanical resistance, and crystallographic phases as a function of the concentration of ZrO_2_ nanoparticles [39]. As observed, density and cold crushing strength values increased with nano-ZrO_2_ addition. Likewise, the mechanical resistance reaches an improvement of almost three times with 5 wt.% of nano-ZrO_2_ addition (315 MPa) compared to the MgO refractory (119 MPa).

As known, the cold crushing strength of a refractory material is related to some other properties which are a direct result of the mechanical resistance, such as abrasion/erosion and slag attack resistance. The stronger a refractory material is, the greater is the abrasion resistance. In addition, stronger refractories are expected to have a higher resistance to slag attacks. Based on the physical, mechanical, and crystallographic properties developed by the 5 wt.% of nano-ZrO_2_ addition in the MgO matrix, this specific concentration was selected to evaluate the chemical resistance. Within the Table 1, CaO with 1.5 appears as an impurity, from raw material, marked with (*).

Copper slag (Atlantic Copper) was used to perform the corrosion test, i.e., to determine the chemical resistance of the sintered refractory samples against the slag penetration. Sintered samples analyzed in this chemical test were 25 mm in diameter and 6 mm in height. Inner holes with a size of ~3.5 mm in diameter with 2.5 mm in depth were drilled into the sintered samples to make crucibles. Then, 5 g slag was placed into the crucibles. Subsequently, the crucibles were fired at 1450 °C for 4 h in an electric furnace (Lindberg/Blue M, 240 V and 30 mA under air atmosphere, located in Escuela Politecnica de Mieres, Asturias, Spain). Both the heating and cooling rates were 5 °C/min. Afterward, the crucibles were then cut using a diamond disc along the centerline after natural cooling. After cutting, surfaces of interest were polished with SiC papers for microscopy evaluation. Cross-sections of polished sintered samples before and after the chemical test were characterized by the scanning electron microscopy technique. The corrosion analysis was carried out by JEOL-6610LV SE, Akishima, Tyo, Japan, equipment using backscattered electron (BSE) mode with a working voltage set of 20 kV. Besides, semiquantitative analyses were performed by an energy disperse X-ray spectroscopy (EDX) detector (Inca energy-200) (located in, Servicios Científicos Técnicos, Universidad de Oviedo, Oviedo, Asturias, Spain). Measurements of the penetration distance were carried out through micrographs (SEM analysis) of the corroded samples. Likewise, slag concentration inside the sample was detected by EDX microanalysis, identifying the variation in the chemical composition through the cross-section of the sample (from the base to the slag powder deposit).

The chemical composition of copper slag and MgO were determined by an X-ray fluorescence (XRF) spectrometer (Axios, PANalytical, located in Universidad Autonoma de Nuevo Leon, San Nicolas de los Garza, Nuevo Leon, Mexico) with an Rh-anode X-ray tube with a maximum power of 4 kW. The phase composition of copper slag and MgO were determined using a Bruker, D8 advance model X-ray powder diffractometer with CuKα radiation (λ = 1.5406 Å) operated at 40 kV and 30 mA (located in, Servicios Científicos Técnicos, Universidad de Oviedo, Oviedo, Asturias, Spain). The scans were performed in the 2θ range from 10 to 90°. A step scan of 0.05 and 1.5 s per step in a continuous mode was used for the 325 mesh-sieved powders from the milled samples.

Table 2 collects the values of the chemical composition and phase percentage of the copper slag. Copper slag is mainly composed of fayalite (Fe_2_SiO_4_) and magnetite (Fe_3_O_4_), as well as copper oxide and sulfide (0.5–2 wt.% Cu) as secondary phases. Iron is the main element in copper slags, >40% [3,4,5,6,7,8,9]. Data of XRF analysis collected in Table 2 indicate that the copper and iron contents (in wt.%) are 1.84 and 42.82, respectively. This fact is consistent with the mentioned above ranges for copper slags.

## 3. Results and Discussion

Figure 1a shows the crystalline phase composition of the copper slag where fayalite (Fe_2_SiO_4_) is the primary phase, while magnetite (Fe_3_O_4_) and copper as oxide (CuFe_2_O_4_, with Cu^2+^ and Fe^3+^) are secondary phases. Figure 1b shows the XRD patterns of 95 wt.% MgO with 5 wt.% nano-ZrO_2_ and 100 wt.% MgO samples sintered at 1650 °C. For the 100 wt.% MgO samples, the only detected crystalline phase was the magnesia (MgO). Meanwhile, in the 95 wt.% MgO with 5 wt.% nano-ZrO_2_ samples, magnesia as the main crystalline phase, as well as zirconia (ZrO_2_) and calcium zirconate (CaZrO_3_) as secondary phases were identified. Calcium zirconate (CaZrO_3_) is a highly refractory phase with a melting point of about 2300 °C. Figure 1c corresponding to the 100 wt.% MgO microstructure is composed of equiaxial MgO grains (with an average size of 5 µm) and interconnected pores (~26). Meanwhile, Figure 1d corresponding to the 95 wt.% MgO with 5 wt.% nano-ZrO_2_, a denser microstructure composed of MgO grains much higher than 30 µm with closed porosity of about 14% is observed. Nano-zirconia and calcium zirconate were identified in the microstructure between MgO grains. The nano-zirconia grain size permitted a higher reaction with CaO (as an impurity from MgO), leading to an in situ calcium zirconate formation and a filler effect that helps to develop a denser microstructure. All grain size measurements were carried out using an ImageJ software analyzer. As known, zirconia changes its crystalline structure from monoclinic to tetragonal at high temperatures (>1200 °C) [38]. However, magnesium ions can substitute zirconia ions (it can accept up to 10 wt.% MgO) and then stabilize zirconia in the tetragonal phase. This phenomenon permits the development of a refractory body with minor cracks, i.e., lower porosity.

Moreover, the above is related to the values obtained for 100 wt.% MgO and 95 wt.% MgO with 5 wt.% of nano-ZrO_2_, resulting in 2.72 and 3.04 g/cm^3^ in terms of BD, as well as 26.25 and 14.49% in terms of AP, respectively. In general, an increase in the density values was obtained with nano-additions of ZrO_2_. This behavior can be attributed to three reasons:(i)The cold isostatic pressing method contributes to the application of homogeneous forces on the sample. The mixture of micrometric and nanometric powders helps to achieve maximum packing. Nanoparticles fill cavities between MgO microparticles leading to a more efficient sintering process.(ii)A suitable sintering temperature: as is usually recognized, the sintering temperature in refractory materials is about 1/2 to 2/3 of the melting point. Particularly, in the present study, a sintering temperature of 1650 °C (0.589 of the magnesia melting point) was used. This temperature promotes higher densification.(iii)The sintering mechanism: an ionic migration between the Zr, which has a valence of 4+, and Mg, which has a valence of 2+. Cationic magnesium vacancies in the cubic structure of MgO are generated due to the migration of magnesium ions when they replace zirconia ions. Considering that cationic vacancies are formed in the periphery of the grain boundary of magnesium oxide, and these must be replaced with other MgO cations within the crystal volume, the ionic migration takes place towards the grain boundaries at high temperatures. Then, the grain boundaries begin to move, joining each other, generating densification of the sample with larger grains and contraction of the material, causing the closure of the pores.

### 3.1. 100 wt.% MgO Sample Tested with Copper Slag

Figure 2 corresponds to the corrosion test by copper slag of the sample with 100 wt.% MgO. Figure 2a shows the sample’s upper area (hole where the slag was placed). In this zone, MgO grains are covered with a layer of copper slag. Figure 2b corresponds to the sample’s middle area, where the slag mainly began to infiltrate into the material by the sample porosity. Corrosion was also evidenced in the magnesia grains, which were chemically attacked by grain boundaries. Figure 2c shows a microstructure corresponding to bottom-area of the sample, MgO grains without being chemically attacked.

Chemical elements analysis by punctual semiquantitative method was carried out (through the sample height) to evaluate the infiltration depth of the slag in the sample (see, Figure 2d). A graph that only considered the chemicals elements that belong to the copper slag is plotted in Figure 2d. The hole depth is ~2000 µm, so the remaining 4000 µm was the possible reaction zone, where infiltration deep was measured. From 0 to 4000 µm, the concentration path of elements other than those of the matrix was measured.

Figure 2d shows the concentration of the slag’s main elements (Fe, Si, Ca, Al, written from highest to lowest concentration) as a function of penetration depth in the sample. At the bottom of the specimen (z = 0), concentration in Fe, Si, Ca, and Al <5% was observed from 0 to 3200 µm. At 2000 µm, a peak, where the concentration rose to 18%, due to internal porosity within the material, although the slag’s infiltration is not significant in this zone. Between 3201 and 4000 µm (surface in contact with the slag), slag infiltration is evident (grows from 5% in 3200 µm to 100% in 4000 µm). From 4001 to 6000 µm, the mean concentration was 35.13%. In this case, the slag penetrated through the hole’s wall, remaining only this concentration of slag in the hole.

Figure 2e shows the cross-section of the sample where the SEM micrographs were taken (letters A, B, and C correspond to Figure 2a–c), as well as the slag elements concentration profile represented in Figure 2d obtained by EDX (dotted red line). Figure 2e schematically shows the sample used in the corrosion test. In the hole periphery, the corrosion mechanism is shown. MgO grains were firstly attacked by copper slag through the porosity and then by the grain boundaries (interpreted by dark lines in the scheme).

In 2000 µm, a high amount (18%) of slag elements as (mainly Ca) was found in the refractory matrix, which means that the slag infiltration reached this particular zone. The distance from the bottom of the hole to the last infiltrated area was ~2000 µm.

From 4001 to 6000 µm (the hole area), the slag elements’ average concentration detected was 35.13%.

As observed, Si is one of the chemical elements detected by EDS analysis that might lead to the forsterite phase (Mg_2_SiO_4_) formation. Forsterite phase formation, which will expand and cause forsterite bursting, is one of the main reasons for the failure of refractories [10]. The forsterite phase will form at the beginning of the reaction. Although forsterite has a high melting point (1890 °C), the substitution of MgO by FeO originating from the copper smelting slag will gradually decrease the melting point of the olivine crystal. After sufficient reaction time with the slag, the melting point of the low-MgO-containing olivine [2(Fe, Mg)O SiO_2_] solid solution will approach or even drop below the operating temperature of the smelting furnace [40]. The solid olivine will then be dissolved into the slag and gradually peeled off the refractory. With the destruction of the periclase bonding, the magnesia grains will no longer be bonded together.

According to the SEM and EDX results, monocalcium ferrite (CaFe_2_O_4_) and dicalcium ferrite (Ca_2_Fe_2_O_5_) could be presented in the slag infiltrated zone. In the reported literature, MgO reacts as follows with calcium ferrite [41]:MgO + 2CaFe_2_O_4_ → MgFe_2_O_4_ + Ca_2_Fe_2_O_5_(1)

The low viscosity of both Ca-ferrites and the high wettability of MgO particles by these compounds can result in low penetration resistance. Compared to calcium ferrite slag, the ferrous calcium silicate slag is less aggressive and penetrates the brick less. This phenomenon is attributed to the higher viscosity and lower surface tension of the ferrous calcium silicate slag [41].

On the other hand, CuO can react severely with MgO from the refractory forming guggenite (Cu_2_MgO_3_). The following reaction between CuO and MgO can take place:MgO + 2CuO → Cu_2_MgO_3_(2)

This corrosion experiment demonstrates no formation of guggenite from CuO and MgO since CuO was not detected in the EDX analysis. The absence of CuO is an advantage since, within a short period, the bonding periclase phase can completely be consumed by the guggenite. According to the MgO–CuO phase diagram [42], Cu_2_MgO_3_ will decompose back to Cu_2_O and MgO at temperatures above 1062 °C. This phenomenon suggests that the guggenite phase inside the refractory will decompose at temperatures above 1062 °C, causing spalling of the refractory structure if the refractory is reheated [19].

### 3.2. 95 wt.% MgO with 5 wt.% ZrO_2_ Sample Tested with Copper Slag

Figure 3 corresponds to the corrosion test with copper slag of the sample composed of 95 wt.% MgO with 5 wt.% nano-ZrO_2_ addition. Figure 3a shows a micrograph corresponding to the sample’s upper area, where the slag was deposited into the hole. Figure 3b shows the area where the slag begins to infiltrate into the material through the porosity. Likewise, in Figure 3b, slag advances in two different ways: (i) direct infiltration of slag particles through porosity (with a penetration depth of 25 µm from the hole). This penetration depth is less than in the 100 wt.% MgO sample. (ii) When the particles were inside the material, they dissolved in the form of molten slag (liquid) through the grain boundaries (the slag circulates through intergranular spaces).

It is assumed that this refractory matrix has three main advantages, which help to increase the chemical attack resistance of MgO grains against the copper slag.

(i)CaZrO_3_ in situ formation helps to hinder the slag penetration in the refractory by increasing the viscosity of the molten slag, slowing down the intergranular path of the molten slag.(ii)The slag infiltration was inhibited due to the quasi-spherical ZrO_2_ nanoparticles since they act as barriers against the penetration of the intergranular liquid. Retention points were observed as non-circular phases due to accumulation in different proportions of slag elements on the periphery of ZrO_2_ nanoparticles, where the slag was retained around the ZrO_2_ particles.(iii)This matrix exhibited a higher density and lower porosity which also help to avoid a considerable slag infiltration.

Figure 3c shows the bottom area of the sample, where it is possible to see ZrO_2_ particles embedded in the MgO matrix with no slag penetration.

A punctual semi-quantitative analysis of elements was conducted to analyze the slag infiltration distance through the sample (see Figure 3d). In the graph, only chemical elements that belonged to the copper slag are plotted. For the infiltration measurement, the height of 3617 µm was considered as a reference since the hole measured ~2383 µm. Starting from the reference height downwards, the concentration path of chemical elements other than those of the matrix was measured.

Figure 3d shows the profile of the concentration of slag elements (Al, Si, Ca, Ti, Fe, written from highest to lowest concentration) as a function of the height of the sample (slag penetration depth, z = 0 at the bottom of the sample). It is possible to detect three different zones: from 0 to 2981 µm (there is not slag infiltration); between 2982 and 3617 µm (the concentration grows from 0 to 51.5%); finally, from 3618 to 6000 µm (the mean concentration was 55.3%). In this case, the slag remained in greater quantity in the place where it was deposited (less infiltration than in the previously studied case).

Letters A, B, and C correspond to Figure 3a–c, as well as the slag profile concentration-distance represented in Figure 3d obtained by EDX microanalysis (dotted red line). Figure 3e schematically shows the sample’s cross-section used in the corrosion test where the SEM micrographs were taken. Dark lines represent the corrosion path, where the MgO grains were attacked by copper slag through the porosity and grain boundaries.

At 2982 µm, Al, Si, Ca, Ti, and Fe slag elements were found in the refractory matrix, which means that the slag infiltration reached this zone. The distance from the bottom of the hole to the last infiltrated area was ~636 µm.

A specific study was also carried out in the hole section, i.e., from 3618 to 6000 µm. The average concentration detected in this area was 55.3%. That fact indicates that in the sample with 5 wt.% nano-ZrO_2_ addition, remained a greater quantity of slag than the 100 wt.% MgO sample. The last might indicate a higher chemical resistance against copper slag.

Monocalcium ferrite, dicalcium ferrite, and forsterite phases might be formed in the slag infiltrated zone. Nano-zirconia can interact with these phases. Therefore, the high reactivity of nano-zirconia, associated with the formation of a higher melting system and/or higher slag viscosity, prevents the penetration of the liquid phases deep into the brick.

It has also been reported that calcium zirconate can modify the liquid phase’s viscosity. The formation of higher slag viscosity might block the penetration of the slag into the brick [43,44,45].

It is proposed that SiO_2_ might react with ZrO_2_ (from the refractory) to form the zircon phase (ZrSiO_4_). Zircon is recognized as a high chemical resistance phase. The following reaction between SiO_2_ and ZrO_2_ can take place:ZrO_2_ + SiO_2_ → ZrSiO_4_(3)

Figure 4 shows the microstructure corresponding to the 100 wt.% of MgO and 95 wt.% of MgO with 5 wt.% of nano-ZrO_2_ samples chemically attacked by copper slag. The microstructural analysis was carried out at the middle of the sample (where the deposited copper slag is located). In both samples, the infiltration was mainly through the porosities. In Figure 4a,b is observed magnesia grains chemically attacked by the slag where the advance of slag was through porosity and grain boundaries. The ZrO_2_ nanoparticles are located mainly in the triple points. They act as retention points of the slag, preventing the advance of the liquid phase. The in situ CaZrO_3_ also contributes to slow down the slag infiltration, increasing the viscosity of the slag.

## 4. Conclusions

In both systems, the slag infiltration was mainly through the sample’s porosity. According to the analysis, in the 100 wt.% MgO sample with 26.24% of porosity, the penetration of copper slag (2000 µm) was higher than the 95 wt.% MgO with 5 wt.% nano-ZrO_2_ addition sample with 14.48% of porosity (~636 µm).

According to the SEM and EDX results, monocalcium ferrite (CaFe_2_O_4_), dicalcium ferrite (Ca_2_Fe_2_O_5_), and forsterite (Mg_2_SiO_4_) phases might be formed in the slag infiltrated zone in both refractory sample (100 wt.% of MgO and 95 wt.% of MgO with 5 wt.% of nano-ZrO_2_).

The low viscosity of both Ca-ferrites and the high wettability of MgO particles by these compounds can result in low penetration resistance of 100 wt.% MgO refractory sample. Meanwhile, for the 95 wt.% of MgO with 5 wt.% of nano-ZrO_2_ addition, the high reactivity of nano-zirconia, associated with the formation of a higher melting system and/or higher slag viscosity, prevents the penetration of the liquid phases deep into the brick.

The ZrO_2_ nanoparticles are located mainly in the triple points. They act as retention points of the slag, preventing the advance of the liquid phase. The in situ CaZrO_3_ also contributes to slow down the slag infiltration, increasing the viscosity of the slag.

It is proposed that SiO_2_ might react with ZrO_2_ (from the refractory) to form the zircon phase (ZrSiO_4_). Zircon is recognized as a high chemical resistance phase.

The evaluation presented gives the primary results and shows the corrosion wear process with the MgO sample and with the nano-ZrO_2_ sample, which serves to provide basic knowledge for the replacement of magnesia-chromite refractory materials in the copper industry.

## Figures and Tables

**Figure 1 materials-14-02277-f001:**
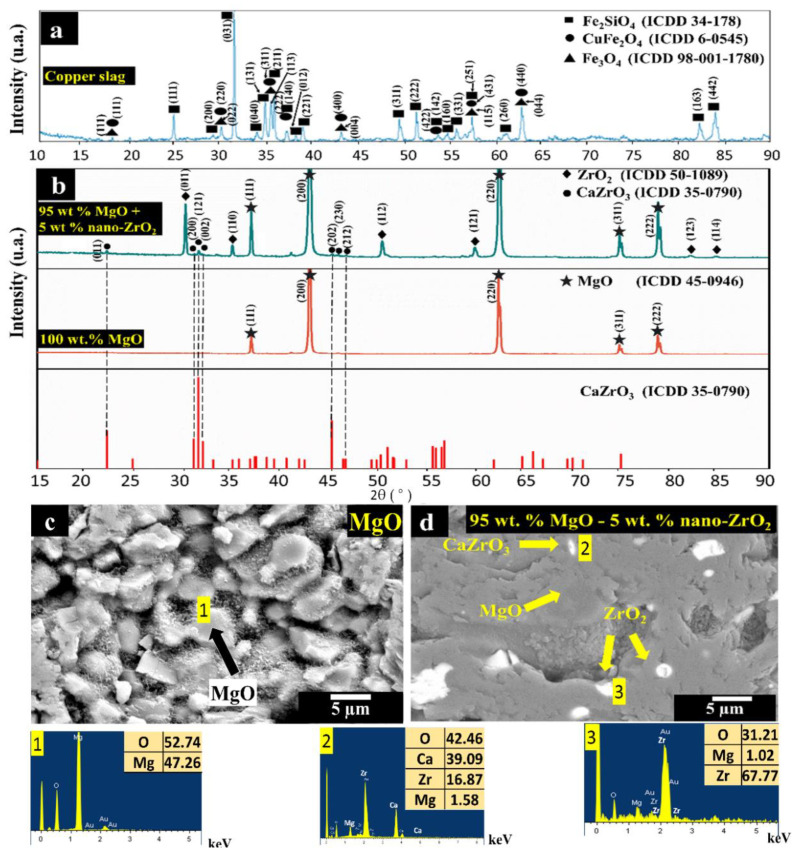
(**a**) XRD pattern of copper slag; (**b**) XRD pattern of samples with 95 wt.% MgO with 5 wt.% nano-ZrO_2_ and 100 wt.% MgO; (**c**,**d**) SEM images of samples of 100 wt.% MgO and 95 wt.% MgO with 5 wt.% nano-ZrO_2_, respectively. Within Figure (**c**,**d**), phases were detected which were performed an EDX analysis, represented by the numbers 1–4.

**Figure 2 materials-14-02277-f002:**
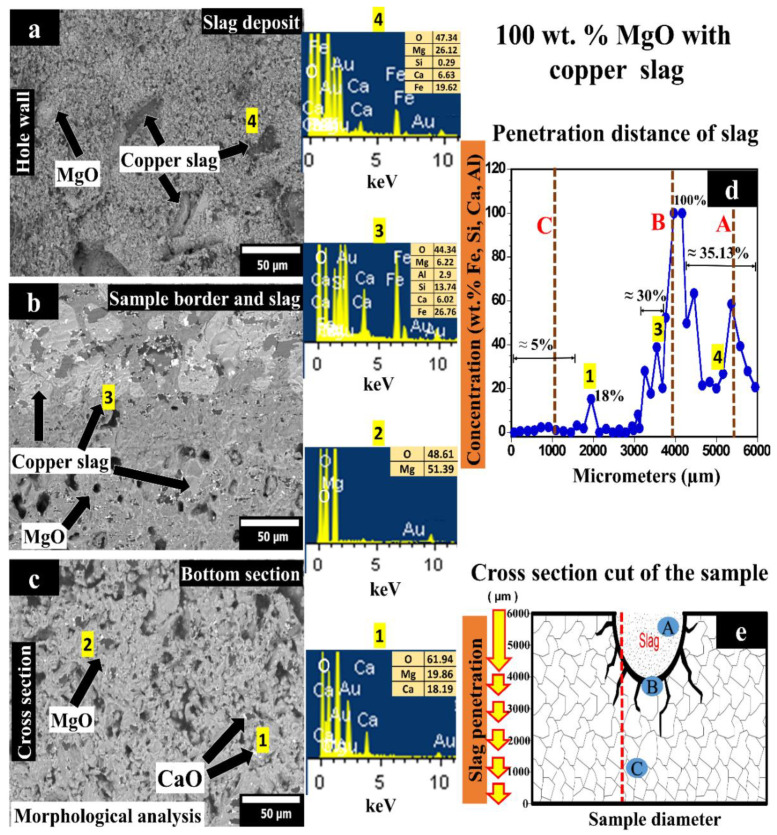
(**a**–**c**) SEM images of sample with 100 wt.% MgO tested with copper slag; (**d**) concentration of slag elements as a function of the penetration distance; (**e**) schematic representation where SEM (dots with letters) and EDX analyses (red line) were made. Within Figure (**a**–**c**), phases were detected which were performed an EDX analysis, represented by the numbers 1–4.

**Figure 3 materials-14-02277-f003:**
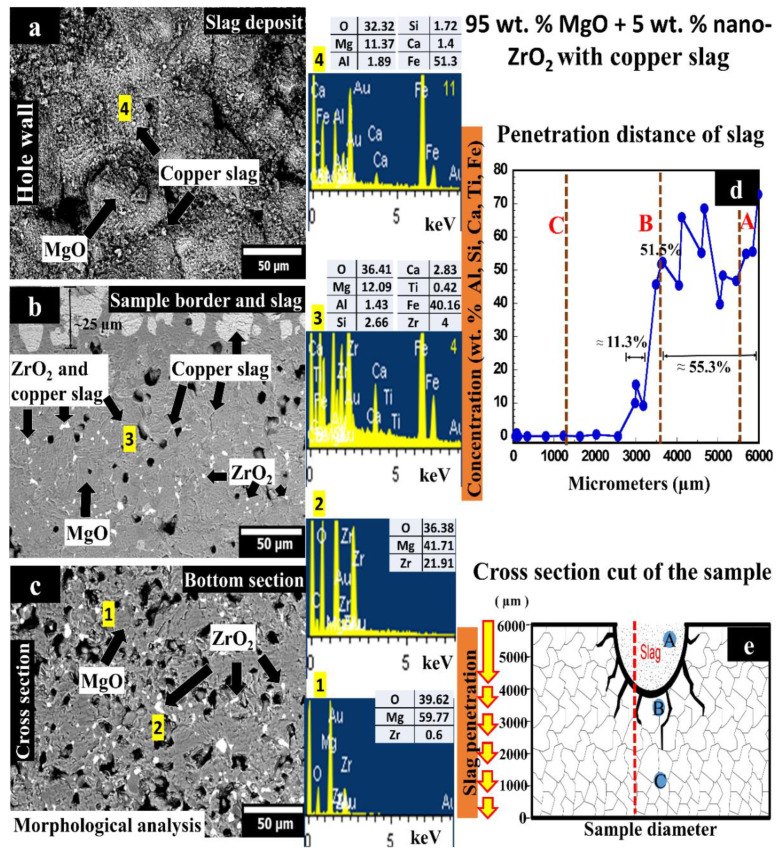
(**a**–**c**) SEM images of sample with 95 wt.% MgO with 5 wt.% nano-ZrO_2_ tested with coper slag; (**d**) concentration of slag elements depending on the penetration distance; (**e**) schematic representation where SEM (dots with letters) and EDX analyses (red line) were made. Within Figure (**a**–**c**), phases were detected which were performed an EDX analysis, represented by the numbers 1–4.

**Figure 4 materials-14-02277-f004:**
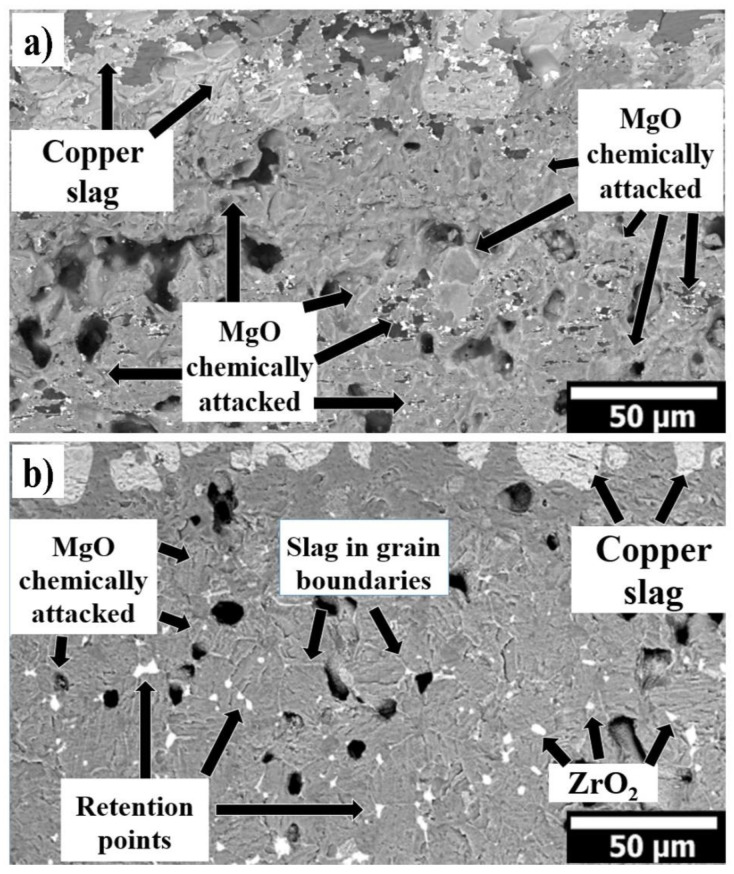
(**a**) SEM images of sample with 100 wt.% MgO tested with coper slag; (**b**) SEM images of sample with 95 wt.% MgO with 5 wt.% nano-ZrO_2_ tested with coper slag.

**Table 1 materials-14-02277-t001:** Physical and mechanical properties as a function of ZrO_2_ nano-additions of samples sintered at 1650 °C, pressed by cold isostatic pressing. The intensity of the phases found in each specimen is represented by (X).

1650 °C					
Cold isostatic pressing	**Properties**	**Composition MgO + (x = wt.%) Nano-ZrO_2_**			*** Impurity from MgO, CaO ~ 1.5**
		**x = 0**	**x = 1**	**x = 3**	**x = 5**
	Density (g/cm^3^) [39]	2.72	2.87	2.99	3.04
	Porosity (%) [39]	26.24	20.56	16.43	14048
	Cold crushing strenght (MPa)	119	210.07	304.72	315.78
	Crystallographic phases	MgO X X X	MgO X X X	MgO X X X	MgO X X X
			ZrO_2_ X	ZrO_2_ X X	ZrO_2_ X X X
		-	CaZrO_3_	CaZrO_3_ X	CaZrO_3_ X X

**Table 2 materials-14-02277-t002:** Quantitative analysis of elements and phases in copper slag (wt.%).

Copper Slag			
Element/wt.%	Element/wt.%	Element/wt.%	Element/wt.%
Ca/1.756	Cr/0.02603	Al/3.239	Cu/1.84
O/36.15	Cl/0.02477	P/0.05097	Co/0.4598
Fe/42.82	K/0.6914	Ti/0.1994	Mo/0.1796
Si/10.36	Sr/0.01129	S/0.2197	As/0.1535
Mn/0.1327	Zn/0.2614	Na/0.993	Pb/0.1129
**Phases (%)**			
Fe_2_SiO_4_ = 85.80 ± 1.30		Fe_3_O_4_ = 7.90 ± 1.60	CuFe_2_O_4_ = 6.20 ± 1.80

## Data Availability

The data presented in this study are available on request from the corresponding author.

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
