# Peer review of "Research and Development of Novel Refractory of MgO Doped with ZrO2 Nanoparticles for Copper Slag Resistance"

_materials, 2021, doi:10.3390/ma14092277_

Round 1
Reviewer 1 Report
The manuscript deals with interaction between copper slag and MgO-ZrO2 ceramic composites. The manuscript contains rather limited experimental data (just two samples), but a long introduction. I highly recommend to be more specific in Introduction. Please follow the remarks:
-Please explain in detail the selected concentration of the ZrO2 additive. How it affects pressing and sintering?
-Fig. 1c,d How did authors identify phases? At least EDX mapping should be added.
-Lines 155 - 164. These assumptions should be supported by some data. What about porosity of the samples and other sintering mechanisms?
-Fig. 2a-c, and Fig. 3a-c. Again, phase identification is questionable without EDX mapping. Readers even do not know the SEM mode, type of detector, voltage and so on.
-Fig. 2e and Fig. 3e. Do these cracks on the scheme mean anything? Do we know anything about diffusion of the slag? It seems nothing.
-What is the influence of ZrO2 addition on mechanical properties?
-What kind of corrosion is mentioned in the Conclusion?
Author Response
Dear reviewer, The answers are in the attached document.

Reviewer 2 Report
This manuscript reports a study on the corrosion behavior of a MgO-5 wt. % nano ZrO2 composite compared to the 100 wt. % MgO base-case when both materials are in contact with copper slags. The authors claim that the composite shows higher corrosion resistance. Basically, this conclusion is gained from the comparison between Figure 2d) and Figure 3d), where the penetration distance is shown for both systems. My opinion is that this manuscript presents some flaws:
1) The compared materials shown a different infiltration behavior, which cannot be automatically related to the corrosion resistance;
2) Infiltration strongly depends on materials porosity. The compared materials have different relative density. Therefore, it is likely that the observed differences in infiltration are due to the different porosity, and not to the addition of nano ZrO2, as claimed by the authors;
3) The authors do not explain why they used nano ZrO2, in place of, for instance, micrometer ZrO2.
Accordingly, I do not think this manuscript should published since the conclusions are not supported by the reported results.
Author Response
Dear Reviewer
The answers are attached in the document.

Reviewer 3 Report
My comments:
1. Lines 56-59. It is said "As it is known, matte and slag phases attack the refractory lining at a high temperature in a detrimental way, increasing the deterioration in the slag line, where bricks are mainly of MgO- Chromite, MgO- Al2O3, MgO-CaO that face the serious problem of the corrosion by molten slag, which is worsened by the presence of high levels of dissolved cuprite, a highly effective flux.". It is known, please give references proved this statement in detail.
2. The experimental description lacks a few important details: companies delivering MgO and chemicals, types of furnace, sintering equipment, EDX.
3. It is written (from line 171): "the concentration of the main elements of the slag (Fe, Si, Ca, Al, written from highest to lowest concentration) as a function of penetration depth in the sample (z=0 at the bottom of the
specimen): concentration in Fe, Si, Ca and Al< 5% was observed from 0 (bottom of the sample) to 3200 µm, with a peak at 2000 µm, where concentration rose up to 18%, due to internal porosity within the material, although the infiltration of the slag is not significant in this zone; between 3201 µm and 4000 µm (surface in contact with the slag), slag infiltration is evident (grows from 5% in 3200 µm to 100% in 4000 µm), from 4001 µm to 6000 um". And further, "total penetration of copper slag in the 100 wt. % MgO sample was ~ 2000 µm, the mean concentration was 35.13%,". So, what is a real corrosion depth?
4. And the same below (from line 211): "profile of concentration of slag elements (Al, Si, Ca, Ti, Fe, written from highest to lowest concentration) as a function of the height of the sample (slag penetration depth, z=0 at the bottom of the sample). It is possible to distinguish three different zones: from 0 to 2981 µm, there is not slag infiltration; between 2982 µm and 3617 µm the concentration grows from 0 to 51.5%; finally, from 3618 µm to 6000 µm, the mean concentration was 55.3%". And below: "In the case of
the 95 wt. % MgO with 5 wt. % nano-ZrO2 sample, the penetration of slag was ~636 µm, which is significantly smaller than in the 100 wt. % MgO sample.". Again and as above, both parts need rewritting and explanation to a reader what mean the values in the first part and how the slag penetration has been measured. On the other hand, what is a correlation between slag penetration and elemental profiles.
5. Line 232: "in the sample with 100 wt. % MgO, the penetration of copper slag (3000 µm)"; earlier 2000 um has been reported.
Author Response

(The authors gave the same response as above.)

Reviewer 4 Report
The authors investigated copper-slag corrosion of a nano-ZrO2 aided MgO refractory in comparison to a pure MgO one. The topic is introduced very well and the experimental findings are precisely described. However, some points should presented in a better way before the manuscript can be accepted for publication.
Please see the following comments and suggestions:
l97: Please explain in more details why ZrO2 was added to the material. Which chemical reactions can be expected in contact with the copper slag from phase diagram?
l142: Which phase shows grain growth? I guess you mean the MgO grains. How did you quantify this observation?
l158-164: Another reason for densification could be the addition of a finer material (ZrO2) acting as filler of pores between MgO grains.
figure 2: The corrosion process you explain in l165-186 can be hardly seen in figures 2a-c. I suggest to show additional micrographs with larger magnification as the the ones now are more less meaningless regarding your explanation. The same can be said for figures 3a-c.
l228, conclusions: How does the different porosity influence the penetration of the slag into the refractory material?
Author Response

(The authors gave the same response as above.)

Round 2
Reviewer 1 Report
I believe the authors have been improved the manuscript and provided extensive comments regarding reviewers remarks.
Reviewer 2 Report
The authors significantly modified the manuscript trying to address my previous comments. They also added several information, although not all them are needed, consistently reported, or well organized.
While my previous comments #1 and #3 can be considered, at least partially, addressed, I still have doubts about the issue I mentioned in my comment #2:
- Infiltration strongly depends on materials porosity.
The compared materials have different relative density. Therefore, it is likely that the observed differences in infiltration are due to the different porosity, and not to the addition of nano ZrO2, as claimed by the authors.
The authors replied as follows:
Thank you for your observations. We have considered an extensive explanation to clarify this point. Now in the manuscript, it is commented that the infiltration of both samples was through porosity since it presented different porosities, but mechanisms that helped to stop the infiltration and protect the chemically attacked MgO grains was:
- CaZrO3 in situ formation helps to hinder the slag penetration in the refractory by increasing the viscosity of the molten slag, slowing down the intergranular path of the molten slag.
- The slag infiltration was inhibited due to the quasi-spherical ZrO2 nanoparticles since they act as barriers against the penetration of the intergranular liquid. Retention points were observed as non-circular phases due to accumulation in different proportions of slag elements on the periphery of ZrO2 nanoparticles, where the slag was retained around the ZrO2particles.
- This matrix exhibited a higher density and lower porosity which also help to avoid a considerable slag infiltration.
While all the mechanisms mentioned above may be certainly reasonable, the relative importance is questionable. The results presented in the manuscript did not give any insights about this topic. In addition, while there is a significant difference in porosity between the 100% MgO sample and the 95% MgO-5% ZrO2 one (cf. Table 1), such that mechanism iii) surely plays an important role, the actual contribution of mechanisms i) e ii) can be only hypothesized. In addition, mechanism i) is due to the presence of CaO as an impurity in the MgO starting materials. Therefore, it seems to me that such an important effect related to an impurity does not sound so much realistic.
As a conclusion, we should recall that novelty of this work relies in the addition of nano ZrO2 to MgO. However, none of the reported results proved that the proposed material would show better corrosion resistance than, for instance, full-dense MgO.
Accordingly, I am regret I have to confirm my previuos opinion: this manuscript should not be published since the claims are not supported by the reported results.
This manuscript is a resubmission of an earlier submission. The following is a list of the peer review reports and author responses from that submission.